# Modeling Highlighting of Metaphors in Multitask Contrastive Learning Paradigms

**Meghdut Sengupta**[1]    **Milad Alshomary**[1]
**Ingrid Scharlau**[2]    **Henning Wachsmuth**[1]
[1]Leibniz Universität Hannover    [2]Paderborn University
[1]{m.sengupta, m.alshomary, h.wachsmuth}@ai.uni-hannover.de
[2]ingrid.scharlau@uni-paderborn.de

## Abstract

Metaphorical language, such as "spending time together", projects meaning from a source domain (here, *money*) to a target domain (*time*). Thereby, it highlights certain aspects of the target domain, such as the *effort* behind the time investment. Highlighting aspects with metaphors (while hiding others) bridges the two domains and is the core of metaphorical meaning construction. For metaphor interpretation, linguistic theories stress that identifying the highlighted aspects is important for a better understanding of metaphors. However, metaphor research in NLP has not yet dealt with the phenomenon of highlighting. In this paper, we introduce the task of identifying the main aspect highlighted in a metaphorical sentence. Given the inherent interaction of source domains and highlighted aspects, we propose two multitask approaches - a joint learning approach and a continual learning approach - based on a finetuned contrastive learning model to jointly predict highlighted aspects and source domains. We further investigate whether (predicted) information about a source domain leads to better performance in predicting the highlighted aspects, and vice versa. Our experiments on an existing corpus suggest that, with the corresponding information, the performance to predict the other improves in terms of model accuracy in predicting highlighted aspects and source domains notably compared to the single-task baselines.

## 1 Introduction

A metaphor can be defined as a cross-domain conceptual mapping from a source domain to a target domain (Lakoff and Johnson, 2003). The abundance in which metaphors occur in everyday language (Gibbs, 1992; Ortony, 1993), such as "*winning* someone's heart" or "tax *evasion*", presents a need for metaphor interpretation and, to that end, the computational decoding of metaphors. Much like in other forms of figurative language, such

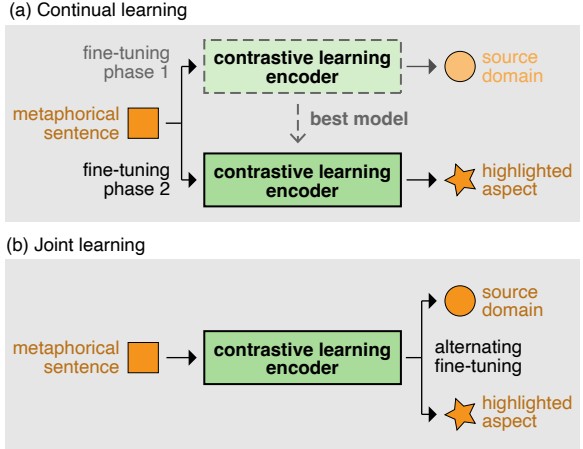

Figure 1: Proposed multitask approaches to predicting the source domain and highlighted aspect of a metaphorical sentence: (a) A contrastive learning encoder is fine-tuned continually, first for source domains and then for highlighted aspects (or vice versa). (b) The encoder is fine-tuned jointly on both in an alternating fashion.

as sarcasm, a central challenge with metaphors is that the implicit intended meaning differs from the meaning of the explicit metaphorical expression (Goatly, 1997; Allott and Textor, 2022). The context provided by the neighboring words in the sentence is an indicator of this implicit meaning (Stern, 2000). However, for better comprehending the meaning manifested by a metaphor in a given context, further levels of understanding are needed; for example, information about the source and target domain (Lakoff, 1987; Johnson and Pascual-Leone, 1989; Robins and Mayer, 2000). Consider the following metaphorical sentence:

*"Excessive tax is killing American family business"*

Here, the word "killing" is used as a metaphor, since *business* as a concept cannot be killed by *tax* in physical realms. The intended meaning is manifested by establishing the mapping of two conceptual domains, drawing the meaning from the source domain (*physical harm*) and projecting the

meaning into the target domain (*taxation*). In this metaphorical context, the word *killing* is the *literal* representative of the metaphorical meaning construction. For simplicitly, we henceforth refer to this word as the *literal metaphor*.

According to past research on metaphors (Lakoff and Johnson, 2003; Wolf and Polzenhagen, 2003; Andriessen, 2008; Maxwell, 2015), the core idea of a metaphor is to *highlight* certain aspects of its target domain while *hiding* others; both aspects have largely been disregarded so far in NLP. The more apparent and deliberate of these is highlighting. In the example above, a highlighted aspect may be *threat*, since the *tax* poses a threat in terms of the economic distress it can cause.

Our research builds on the distinction between source and target domains in line with existing research on computational metaphor interpretation (Stowe et al., 2021). Beyond prior work, however, we argue based on aforementioned linguistic theories that it is also important to have an understanding of the aspects highlighted by a metaphor in a given context for better metaphor interpretation. To fill this gap, we provide the following contributions in the paper at hand:

- We assess for the first time how and to what extent the aspects highlighted by a metaphor can be predicted computationally. In particular, we study the hypothesis that, by concurrently considering source domain and highlighted aspects and by effectively exchanging information between them, we can enhance their identification. In simpler terms, we investigate whether a joint modeling of source domains and highlighted aspects improves their predictability.

- To implement our hypothesis, we develop two multitask contrastive learning approaches to the most highlighted aspect and the source domain in metaphorical sentences as illustrated in Figure 1: one using continual learning, the other using joint learning. We analyze whether, in this setup, involving the information of the highlighted aspects benefits the prediction performance of the model on the source domains, and vice versa.

Given the corpus of Gordon et al. (2015) with metaphorical sentences annotated for highlighted aspects and source domains, we evaluate different variations of our approaches against a single-task baseline for both labels. Our results indicate that, in almost all cases, the continual learning approach outperforms a single-task setup, indicating that the combined information of source domains and highlighted aspects benefit the models to predict either of them. Our analysis of the results suggests that continual learning particularly learns to differentiate between single source domains and a composite source domains well.

## 2   Related Work

Highlighted and hidden aspects of metaphors contribute to the implicit intention conveyed by the metaphor in the given context. As we discuss in the following, different past research has utilized the source and the target domains in the computational analysis of metaphors. To the best of our knowledge, however, no one has investigated how aspects are highlighted by metaphors yet.

Shutova et al. (2013) defined an ideal metaphor processing system for NLP applications to consist of two components: *metaphor detection* and *metaphor interpretation*. In line with their work, Shutova et al. (2012) previously identified metaphors to then model metaphor interpretation as a paraphrasing task (Witteveen and Andrews, 2019). Similarly, Mao et al. (2018) used WordNet (Miller, 1995) to explore the contextual domains of metaphors to then successively identify and interpret metaphors as a use case for machine translation. The authors also design the interpretation step as a paraphrasing task.

The majority of computational research on metaphors in natural language has focused on metaphor identification so far, where the task is usually treated as a binary classification task: Given an input word or sentence, decide whether the meaning it represents is metaphorical or literal (Steen, 2010; Li et al., 2013). One of the pioneering works approached the identification with unsupervised spectral clustering techniques based on relevant parts of speech, such as nouns and verbs (Shutova et al., 2010). Using a seed phrase to learn similar metaphors associated with particular source domains, the authors adapt the conceptual mapping from one domain to the other. More recent approaches include that of Li et al. (2023) which employs FrameNet (Ruppenhofer et al., 2016) to identify metaphors where a RoBERTa-based pretrained language model (Liu et al., 2019) is finetuned to encode contextual cues of the concepts

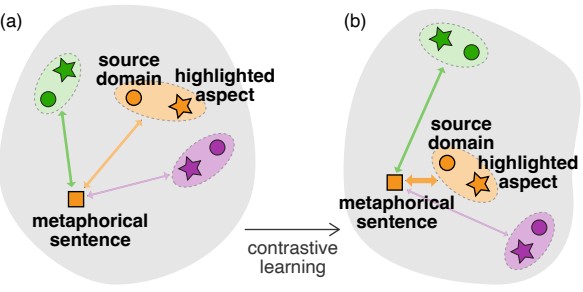

Figure 2: Learning to predict the *source domain* and *highlighted aspect* for a given *metaphorical sentence*: (a) In the original embedding space, the similarity of the sentence to the correct domain and aspect may be not be high enough. (b) In the learned space, their similarity is increased, and it is decreased for others. We exploit the intrinsic semantic similarity of source domain and highlighted aspect through multitask learning.

in the data from FrameNet for enhanced model performance. First they obtain a sentence representation with one encoder. Next, they employ a separate encoder to form FrameNet based representations of the concepts associated with the given literal metaphor. Finally they combine them with the sentence representations to obtain a joint representation, which is passed through a softmax layer for the final prediction. Unlike all these works, our research targets the broader interpretation of metaphors by exploiting components of metaphorical meaning construction, like source domains and highlighted aspects.

Based on the founding work of Lakoff and Johnson (2003) previous work in metaphor interpretation has explored the source and target domains with the usage of FrameNet (Stowe et al., 2021) and in prediction of source domains (Sengupta et al., 2022). In the latter, they have proposed to predict source domains in given metaphorical sentences using contrastive learning (Zhang et al., 2022). We incorporate their contrastive learning idea in our approach and evaluate it on the same metaphor corpus (Gordon et al., 2015). However, not only is our main goal to predict highlighted aspects, but we also exploit the semantic connection between source domains and highlighted aspects using multitask learning.

## 3 Approach

In this section, we present how to predict both source domains and highlighted aspects jointly

with multitask learning. We start from our contrastive learning approach from previous work that we shortly summarize in the following subsection. Then, we propose two alternative multitask schemes for the prediction on this basis: continual learning and joint learning. A comparison of the two variants is shown in Figure 1.

### 3.1 Contrastive Learning for Metaphor Interpretation

As discussed in Section 2, Sengupta et al. (2022) demonstrated that modeling similarities between metaphorical sentences and their corresponding source domain through contrastive learning boosts the effectiveness of trained models on the source domain prediction task. The core idea is to learn an embedding space where the given input sentence representation and the representation of the most similar source domain is trained to be situated close to each other and the less likely source domain to be situated further apart.

For the work at hand, we extend their idea to both source domains and highlighted aspects. In particular, our hypothesis underlying the multitask approaches presented below is that the mutual semantic relations between source domains and highlighted aspects can be modeled in a the learned embedding space for their joint prediction. We illustrate this idea in Figure 2.

### 3.2 Continual Learning

Our first approach follows the idea of *continual learning*, illustrated in Figure 1(a). Research has shown that continual learning can be helpful in a variety of downstream tasks with regard to the performance improvement of machine learning models (Ke et al., 2021).

Fundamentally, continual learning is a machine learning technique where principally a model is trained on different tasks in a sequential order (Hadsell et al., 2020; Scialom et al., 2022b). This process directly preserves the information obtained in the first task and leverages that information in the next task where the model is finetuned (Scialom et al., 2022a).

We adapt this technique to train our models on the task of predicting the source domains and highlighted aspects sequentially. In particular, given a set of metaphorical sentences as input, we first finetune a Sentence-BERT (Reimers and Gurevych, 2019) with DeBERTa (He et al., 2020) as the encoder to learn the most similar source domains.

| Dataset | Metaphorical Sentences | Source Domains | Highlighted Aspects |
|---|---|---|---|
| Training set | 1000 | 120 | 61 |
| Validation set | 128 | 56 | 33 |
| Test set | 301 | 81 | 49 |
| Full corpus | 1429 | 138 | 78 |

Table 1: The distribution of *metaphorical sentences* in our experimental setup along with the numbers of *source domains* and *highlighted aspects* the sentences refer to.

Next, we take the best-performing model from this training phase on the validation set and continue fine-tuning it to predict the most similar highlighted aspects for the same set of sentences. Both training phases employ contrastive learning (as discussed in Section 3.1) by using the multiple-negatives ranking loss (Henderson et al., 2017) like Sengupta et al. (2022) where for a sentence representation **a** and a given correct highlighted aspect **b** for every positive pairs $(a_i, b_i)$, for a negative pair $a_i$ for every highlighted aspect $b_j$, $j \neq i$, if $m = |A| = |B|$ and S is the similarity score computed from the sentence embeddings for the given pairs, the loss is computed as:

$$
\begin{aligned}
&\mathcal{L}(\mathbf{a}, \mathbf{b}) \\
&= -\frac{1}{m} \cdot \sum_{i=1}^{m} \log P_{\text{approx}}(b_i | a_i) \\
&= -\frac{1}{m} \cdot \sum_{i=1}^{m} \left( S(a_i, b_i) - \log \sum_{j=1}^{m} e^{S(a_i, b_j)} \right)
\end{aligned}
$$

This loss function optimizes the embedding space such that the representation of the correct aspect highlighted and the representation of the sentence are positioned closer to each other while distancing the representation of the sentence from the incorrect concept representations as shown in figure 2. Similarly, we employ the same technique the other way round to first learn on highlighted aspects in order to then predict source domains.

### 3.3 Joint Learning

Our second approach follows the idea of *joint learning*, as illustrated in Figure 1(b). The approach employs a joint-learning technique where, at the core, it is trained via contrastive learning optimized with multiple-negatives ranking loss, as described in the previous subsection.

We adapt the multitask training procedure of the Sentence-BERT architecture of Reimers and

| Sentence | Metaphor | Src. Domain |
|---|---|---|
| The sad news is with the exception of very few no firearm organisation is doing anything of the slightest value in fighting gun control. | fighting | Struggle, War |
| This is the historical context of Obama's election victory. | victory | Competition, Game, War |
| They attack ""rich people"" while enjoying all the spoils of their luck, I have zero problems with earned wealth, but these clowns literally lucked out in life. | attack | War |

Table 2: Example sentences from the dataset having one or more than one concepts grouped as the source domain. The table is reused from Sengupta et al. (2022).

Gurevych (2019), where the training phase is optimized in a round-robin way. For a given metaphorical sentence, the loss is first computed for learning the correct source domain representation. Next, it is backpropagated to compute the loss for learning the representation of the highlighted aspect.

Hence, the training happens in an alternating fashion where information about the highlighted aspects and source domains is mutually exploited via a shared encoder with hard parameter sharing (Ruder, 2017). For comparability to the continual learning approach, we employ the same encoder to form the representations in this case as well.

## 4 Data

For our experiments, we employ the corpus of Gordon et al. (2015), which is to our knowledge the only metaphor corpus so far that is annotated for both source domains and highlighted aspects. In the following, we briefly summarize its characteristics and the dataset splits we use.

### 4.1 Corpus

The corpus of Gordon et al. (2015) consists of 1771 metaphorical sentences collected from press releases, news articles, weblog posts, online forum discussions, and social media. Each sentence is annotated for several concepts including the *source domain*, the *target domain*, and the *literal metaphor*. For each given metaphorical sentence, a highlighted aspect is annotated (referred to as *schema slot* in the data). For example, *threat* is an aspect highlighted by some sentences with the source domain *addiction*.

## 4.2 Datasets

We use the same dataset partitions as Sengupta et al. (2022). To obtain a clean experimental setting, the authors combined multiple source domains that were assigned to the same sentence into a single source domain (composite source domains) and removed duplicate sentences in the data to finally have 1429 instances. So for example, if a metaphorical sentence had multiple source domains such as *competition, game, and war* annotated, in their work the combination *competition+game+war* was treated as a single source domain, which was a different source domain from, for example, *competition* as hown in Table 2.

In the case of highlighted aspects, the only combination present is *enemy/side*. For consistency in the task design, we also treat it as a separate label. So, with a 70-30 train-test split, for each of our experiments, we have 1000 training samples, 128 validation samples, and 301 test samples.

Table 1 shows the data distribution in the final corpus for our experiments. As shown in table 1 the experiment corpus has 1429 with 138 source domains and 78 highlighted aspects respectively - emphasizing the sparse label distribution for the downstream tasks of predicting source domains and highlighted aspects. This sparsity is intrinsic to metaphorical language due to its strong diversity.

## 5 Experiments

This section describes setup of the experiments we carried out on the data from Section 4 to study the effectiveness of the two proposed multitask learning approaches from Section 3. We present our basic experimental setup, before we give details on the single-task baseline that we compare to as well as on the two approaches.[1]

**Task Input**    A limitation of the approach of Sengupta et al. (2022) is that it appends the literal metaphor annotated in the corpus of Gordon et al. (2015) to the input metaphorical sentence (with a separator token <SEP> in between).

In a real-world setting this information may not be available. In our experiments, we predict source domains and highlighted aspects *without* the literal metaphor as input. However, to see the effectiveness of our approaches, we also report on the results

*with* the literal metaphor, with the corresponding single-task baseline setups.

**Baseline**    We compare our multitask approaches to a single-task approach, namely we use the contrastive learning approach of Sengupta et al. (2022) here, trained separately for each task.

In this setup, during training, the input sentence is first provided to the encoder in order to obtain the input sentence representation. Similarly, the corresponding label (highlighted aspect or source domain depending on the downstream task) is provided as an input to the same encoder to get the label representations - and hence the corresponding weights are shared.

After that via contrasting learning, the input sentence representation is compared to all the incorrect label representations with paired cosine distance where the training procedure is optimized with multiple negatives ranking loss - which learns the embedding space of the representations such that the given input sentence and the correct label are situated in close proximity in the embedding space.

At inference, the model receives an input sentence in the encoder and forms the representation as stated before. Then it compares the sentence representation with all the labels present in the corpus for the corresponding downstream task, with paired cosine distance, and ranks all the labels in the order of their similarity. Finally, the top-ranked (most similar) label is chosen to be the prediction.

For our experiments with the metaphor appended to the input, we perform the training and inference similarly with the literal metaphor appended to the input sentence with a with a separator token <SEP> in both the cases.

**Contrastive Learning**    Within the contrastive learning setting, we use DeBERTA (He et al., 2020), an enhanced encoder built on top of RoBERTa (Liu et al., 2019), which relies on a disentangled attention mechanism and has shown success in recent NLP research (Zhao et al., 2022). We employ it as the encoder for sentence-transformers in all evaluated model configurations.

**Multitask Learning**    To optimize the models of our multitask learning approaches, we performed a hyperparameter search over batch sizes from $\{4, 8\}$, learning rates in $\{2 \cdot 10^{-5}, 3 \cdot 10^{-5}, 4 \cdot 10^{-5}, 5 \cdot 10^{-5}\}$, and epochs in $\{4, 5, 6\}$.

To find the best checkpoint per experiment, we ran each model on each combination over the hy-

---

[1]The source code of our experiments can be found here: https://github.com/webis-de/EMNLP-23

| Task Input | Approach | Highlighted Aspects | | | Source Domains | | |
|---|---|---|---|---|---|---|---|
| | | Acc@1 | Acc@3 | Acc@5 | Acc@1 | Acc@3 | Acc@5 |
| Metaphorical sentence without literal metaphor | Single-task baseline | 0.524 | **0.767** | 0.831 | 0.488 | **0.718** | **0.797** |
| | Joint learning (ours) | 0.491 | 0.734 | 0.847 | 0.455 | 0.664 | 0.734 |
| | Continual learning (ours) | **0.535** | **0.767** | **0.867** | **0.522** | **0.718** | 0.781 |
| Metaphorical sentence with literal metaphor | Single-task baseline | **0.581** | **0.837** | 0.904 | 0.522 | 0.748 | **0.837** |
| | Joint learning (ours) | 0.508 | 0.800 | **0.910** | 0.508 | 0.730 | 0.827 |
| | Continual learning (ours) | 0.522 | 0.817 | 0.905 | **0.571** | **0.764** | 0.821 |

Table 3: Evaluation of our two multitask learning approaches in predicting highlighted aspects and source domains. Without the ground-truth literal metaphor appended to the metaphorical sentence, *continual learning* outperforms the *single-task baseline* in both tasks. With the metaphor appended to the input, the ground-truth information seems to partly outweigh the impact of multitask learning. The best result per block in each column is marked bold.

perparameters for 20 iterations on the validation set. We then evaluated the optimal configuration in each case on the test set.[2]

**Metrics**  Given that the contrastive learning approach creates a ranking, we evaluate all models in terms of top-1, top-3 and top-5 accuracy.

Here, *top-1 accuracy* means that only the highest-ranked output (that is, a highlighted aspect or a source domain, respectively) is chosen to be correct. In *top-3 accuracy* and *top-5 accuracy*, the output is seen as correct, if it is within the first three and first five ranks, respectively.

## 6 Results

The main results of our experiments are shown in Table 3. Overall, our *continual learning* approach largely outperforms both the *single-task baseline* and the *joint learning* approach in predicting both highlighted aspects and source domains. This suggests that the information of one of the concepts improves model performance on the other.

The limited performance of the joint learning approach may speak for that learning on two tasks concurrently confused the model in some cases rather than helping it to learn both tasks better. The following subsections discuss our results in detail.

### 6.1 Highlighted Aspects

We discuss the results separately for the two task variations: given only the metaphorical sentence as input, and additionally given the literal metaphor.

**Without Literal Metaphor as Input**  Compared to the single-task baseline, the application of continual learning improves the top-1 accuracy by 1.1

points from (0.524 to 0.535). While the top-3 accuracy is the same as the baseline, the top-5 accuracy surpasses it clearly (by 3.6 points), suggesting better overall representational capture in continual learning. A value of 86.7 means that it manages to put the right highlighted aspects into the top-5 in almost 7 out of 8 cases.

**With Literal Metaphor as Input**  Given ground-truth information on the literal metaphor, the impact of multitask learning largely disappears (except for top-5 accuracy, where joint learning is strongest with 0.910). This suggests that the information about the literal metaphor suffices to tackle the task with the straightforward single-task model, while the multitask setting might increase complexity unnecessarily.

### 6.2 Source Domains

Again, we look at the two task variations one after the other.

**Without Literal Metaphor as Input**  In the case of predicting source domains, the continual learning setup outperforms the single-task baseline with an overall top-1 accuracy increment by 3.4 points (0.522 vs. 0.488), while it does not seem to help in the case of top-3 and top-5 accuracies.

**With Literal Metaphor as Input**  Finetuning our model to predict source domains in the setting where literal metaphors are added to the input improves the accuracy by 4.9 points, consistent to the case without the metaphor added to the input. The high gain over the single-task baseline indicates that the direct relationship of the metaphor with its source domain further benefits the learning capabilities of the model.

---

[2]Detailed hyperparameter configurations are provided in the appendix.

# 7 Analysis

To further investigate into our results we looked into the predictions of the two tasks for every experimental setup. In order to do that, we obtained the confusion matrices of the outcomes of our experiments (provided in the Appendix) and observed particularly which labels have an improvement resulting from our continual learning approach.

We also looked into the cases where both single-task baseline and continual learning fail to predict the correct outcome, to have a better idea where the approach can be improved further. We classify our analysis in two parts:

**Improvement**  Incorrect prediction by single-task but correct prediction by continual learning

**No Gain**  Incorrect prediction by both single-task and continual learning

## 7.1 Predicting Highlighted Aspects

In the following, we consolidate our main findings for highlighted aspects with selected examples.[3]

**Improvement**  We primarily observed that the continual learning approach performs better in detecting the aspects of *threat* and *threatened*, among others. For example, for the sentence "Taxation destroys earnings and ability to save/invest...inflation destroys monetary wealth already owned.", the continual learning approach predicts the highlighted aspect correctly to be *threat*, unlike *destruction potential* predicted by the single-task baseline.

Continual learning improves performance on the aspect *barrier* such as in the sentence "This would remove a mountain of taxation from the shoulders of labor.", where it correctly predicts the aspect highlighted to be a *barrier* instead of a *scale*.

These particular outcomes indicate that, with the knowledge of the source domain in this case, the continual learning procedure captures the broader implicit meaning better while the prediction of the single-task setup is possibly more influenced by the word *destroy*.

**No Gain**  Both the single-task baseline and continual learning have room for improvements for the aspects of *change* and *agent*. For instance, for the sentence "But some pro-gun legislation, including the sweeping 'guns everywhere bill that was

---

[3]Detailed outcomes of the experiments on the test set are provided with the code provided.

| Sentence | Metaphor | Single Task | Continual Learning |
|---|---|---|---|
| We cannot allow Texas to go down the big government *pathway*. | pathway | **goal** | movement |
| Gun control *advances* in the Senate, Democrats thank MSM for propaganda April 11, 2013 | advances | **agent** | movement |
| These Are the People Our Money Is *Murdering* in Gaza 22 July 2014 | murdering | **criminal** | victim |

Table 4: Comparisons of outcomes on the test set with the literal metaphor appended to the input sentence, to predict highlighted aspects. These examples, show instances where the single-task baseline predicts the highlighted aspect correctly (denoted in **bold** font) but the continual learning predicts them incorrectly.

signed into law earlier this year by Georgia's Republican Gov. Nathan Deal, has advanced in recent months.", both the approaches predict the highlighted aspect to be *agent* instead of *change*.

Furthermore, for the sentence "The reality is that firearm safety has not meaningfully advanced in the past century.", while the true label was *agent*, the single-task setup predicts *change* and the continual learning predicts *movement*, indicating a confusion regarding the three aspects.

## 7.2 Predicting Source Domains

Analogously to the previous subsection, we here present our findings on source domains.

**Improvement**  As stated in Section 4, we have composite source domains present in the corpus which are combinations of individual source domains that are present across the data.

We observe that continual learning can differentiate better between predicting single and composite source domains. An example is the source domain of *struggle* where for the sentence "How can local governments and civil-society organizations effectively fight poverty and promote social responsibility in countries as diverse as Canada, China and Ghana?", continual learning predicts the source domain of *struggle* correctly instead of *struggle and war* by the single-task baseline.

This holds true for other single and their equiv-

alent composite source domains where in the sentence for example "My point was that governments have killed more people - the OP did not say the US government but the last time I looked the US Govt was in fact a 'government and 'governments have killed far more people then individuals not engaged in the service of 'government." the single-task incorrectly predicts *crime and physical harm* while continual learning predicts the correct source domain *physical harm*.

**No Gain**   Overall, continual learning improves considerably over the single-task approach in predicting the source domains. However both the approaches systematically confused among source domains such as *natural physical force* and *body of water*, as in the sentence *sweep out the old, then when the fresh rain of democracy came, a whole new country would spring up, like mushrooms.* or among *building* or *low location* as in the sentence "This rate is an individual's tax floor.".

In the latter, it might be even challenging for humans to be certain, because the metaphor *tax floor* does indicate that the meaning originates from the concept such as *low location*.

### 7.3   Predicting Highlighted Aspects With Literal Metaphor as Input

One possibly unexpected outcome were the top-1 accuracy results for predicting the highlighted aspects with the metaphor added to the input as shown in table 3. Upon inspection, we found out that the continual learning approach was biased towards the aspect *movement* as shown in Table 4.

In the sentence "We cannot allow Texas to go down the big government pathway", with the literal metaphor *pathway* appended to the input, the continual learning approach predicts the highlighted aspect to be *movement* instead of *goal* which is correctly predicted by the single-task baseline.

Similarly, in the sentence "Gun control advances in the Senate, Democrats thank MSM for propaganda April11, 2013", the continual learning setup predicts the highlighted aspect to be *movement*. It is highly likely that with the addition of the literal metaphor to the input, the continual learning approach fails to capture the broader meaning manifested by the sentence. On the contrary, without the addition of the literal metaphor to the input, in the similar experimental setting, continual learning approach predicts all the aspects correctly, supporting our aforementioned theory.

| Approach | Highl. Aspects | | Source Domains | |
|---|---|---|---|---|
| | Acc@1 | Relaxed | Acc@1 | Relaxed |
| Single-task | 0.524 | 0.548 | 0.488 | 0.714 |
| Joint learning | 0.491 | 0.498 | 0.455 | **0.724** |
| Continual learning | 0.535 | **0.555** | 0.522 | **0.728** |

Table 5: Evaluation of all approaches in the relaxed setting, comparing the *Accuracy@1* from Table 3 to the *relaxed* accuracy. No literal metaphor given as input.

Another interesting example is where the continual learning approach predicts the highlighted aspect to be *victim* instead of *criminal*, as shown in Table 4. While theoretically, it can be argued that both of these aspects are highlighted by the metaphorical usage of *murdering* in the sentence, in the realm of the dataset, the single-task baseline predicts correctly. Given the direct relationship of a source domain to the metaphor, these outcomes intuitively make sense, because of the information regarding source domains already present in this case. However, further experiments are required to understand them better.

### 7.4   Relaxed Top-1 Accuracy

Our fine-grained analysis on the test data revealed that continual learning particularly improves in distinguishing single and composite source domains. However, they also showed that the single-task baseline, in most cases, predicted at least one of the concepts in a composite source domain correctly.

To further investigate the performance of our approach we evaluate our approaches to the baseline with a relaxed top-1 accuracy as shown in Table 5. Unlike our main evaluations, in this setting we consider a prediction to be correct if one of the $n$-combinations in the true label is correct. So if the correct composite source domain for a given metaphorical sentence was a combination of source domain *A* and source domain *B*, we consider it to be a correct prediction if the model predicts only the concept *A* as the source domain of this sentence.

This intuitively also makes sense, given these concepts are combined based on their semantic similarity (Gordon et al., 2015) which means that each of the concepts in a composite source domain is semantically equivalent to the composite source domain. For example, in the sentence "Across the globe, free markets and trade have helped defeat poverty, and taught men and women the habits of liberty", the metaphor *defeat* is a meaning manifestation of the composite source domain *Competition,*

*Game, and War.* In this case, we consider it to be a correct prediction if the model predicts *War* to be the source domain.

In the relaxed accuracy setting, continual learning improves top-1 accuracies for highlighted aspects and source domains. Joint learning is more effective than the baseline for source domains, suggesting that semantic similarity is captured well when considering highlighted aspects, despite the complexity of composite source domains.

Overall, our analysis not only suggests that contrastive learning can also be applied to predict highlighted aspects, but also indicates the performance of continual learning in these two downstream task settings in the consistent performance improvement across the experiments.

## 8   Conclusion

The conceptual mapping from source to target domain in metaphorical meaning manifestation involves the highlighting of certain aspects of the target domain. In this study, we have examined the impact of source domain information on predicting highlighted aspects, and vice versa.

To accomplish this, we have proposed two multi-task learning approaches within a contrastive learning setting: one utilizing continual learning, the other joint learning. We have evaluated the performance of our approaches in comparison to according single-task baselines for predicting either source domains or highlighted aspects.

We have found that continual learning enhances model performance for highlighted aspects and source domains, suggesting mutual improvement. Our fine-grained qualitative analysis further confirms the effectiveness of our approaches across various experimental setups. Moreover, in a more informal yet potentially more applicable assessment, our method outperforms the performance of the single-task baseline, demonstrating beneficial outcomes through joint learning.

We conclude that the aspects highlighted by metaphors can be predicted well in the majority of cases—and even more so a small set of candidate aspects (as suggested by our top-5 accuracy results). We see this as a substantial step towards more comprehensive computational interpretation of metaphors. Following Lakoff and Johnson (2003), metaphors do not only highlight certain aspects of a target domain, they also *hide* others at the same time. Future work should thus pay more atten-

tion to what is *not* put emphasis on by a metaphor, which will naturally bring up additional challenges regarding the interpretation of metaphors.

## Acknowledgment

This work has been supported by the Deutsche Forschungsgemeinschaft (DFG, German Research Foundation) under project number TRR 318/1 2021 – 438445824. We thank the anonymous reviewers for their helpful feedback.

## Limitations

Firstly, one of the major limitations of our work is the nature of our downstream tasks, where from a theoretical standpoint, the number of highlighted aspects for a particular metaphor in a given context are unbounded - which means one cannot say with certainty, if a certain concept is the one and only *correct* aspect for a given metaphorical sentence While we have tackled this problem by treating the data as is and by incorporating a real-world setting for our experimental setup, it is also a *limited* real-world setting that we employ since we take the aspects annotated in the dataset to be the only possible aspects for the metaphorical sentences.

Secondly, while our analysis reveals that our proposed continual learning approach improves model performance, without any level of explainibility it is difficult to say exactly to what degree the information from one task *does* help in the model performance of the other.

Finally, to test the generalizability of our approach it is important to test our models on similar datasets. However, to the best of our knowledge, this is the only dataset which fits our task design.

## Ethical Statement

To the best of our knowledge, we understand that there are no ethical concerns with our paper. We use relatively transparent approaches a publicly available dataset.[4]. To the best of our knowledge, it is unlikely that a potential harm is posed by either the data or our methods.

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

# 9 Appendix

## 9.1 Hyperparameter Configurations

## 9.2 Single-Task Setup

**Highlighted Aspects** Without the literal metaphor added to the input, we find the best-performing checkpoint to result from a batch size of 8, a learning rate of $5 \cdot 10^{-5}$, and 5 epochs. With the literal metaphor, 4 epochs are better while the others are hyperparameters identical.

**Source Domains** Without metaphor added to the input, we find the optimized parameters for the best performing checkpoint to be a learning rate of $5 \cdot 10^{-5}$, batch size of 8, and epochs of 6. With the metaphor added to the input we find the optimized parameters for the best performing checkpoint to be a learning rate of $3 \cdot 10^{-5}$, batch size of 8, a epochs of 5.

## 9.3 Continual-Learning

**Highlighted Aspects** Without metaphor added to the input, in the continual learning approach, based on our hyperparameter search we find the optimized parameters for the best performing checkpoint to be a learning rate of $5 \cdot 10^{-5}$, batch size of 8, and epochs of 6. With the metaphor added to the input we find the optimized parameters for the best performing checkpoint to be a learning rate of $5 \cdot 10^{-5}$, batch size of 8, a epochs of 4.

**Source Domains** Without metaphor added to the input, we find the optimized parameters for the best performing checkpoint to be a learning rate of $4 \cdot 10^{-5}$, batch size of 8, and an epoch of 5. The encoder for the second phase of training is the best performing model from the single-setup of predicting source domains and hence it's optimized with the hyperparameters as mentioned above in section 9.2. With the metaphor added to the input we find the optimized parameters for the best performing checkpoint to be a learning rate of $4 \cdot 10^{-5}$, batch size of 8, a epochs of 5.

## 9.4 Joint-Learning

**Highlighted Aspects** In the joint learning setup, we find the optimized parameters for the best performing checkpoint to be a learning rate of $5 \cdot 10^{-5}$, batch size of 8, and epochs of 6. For the model to identify each of the tasks individually, we add a special token of <hghl> for every sentence for the highlighted aspects and <scm> for the source domains. At inference, we test the best performing model on the test set. With the metaphor added to the input we find the optimized parameters for the best performing checkpoint to be a learning rate of $5 \cdot 10^{-5}$, batch size of 8, a epochs of 6.

**Source Domains** The procedure to fine-tune for this task is the same as mentioned as above, while the only difference being at the inference time, where the instead of highlighted aspects we predict source domains. Without metaphor added to the input, we find the optimized parameters for the

best performing checkpoint to be a learning rate of $5 \cdot 10^{-5}$, batch size of 8, and epochs of 6. With the metaphor added to the input we find the optimized parameters for the best performing checkpoint to be a learning rate of $5 \cdot 10^{-5}$, batch size of 8, a epochs of 5.

### 9.5 Confusion Matrices

Next pages show the confusion matrices for each of the outcomes of the model performances on the test set.

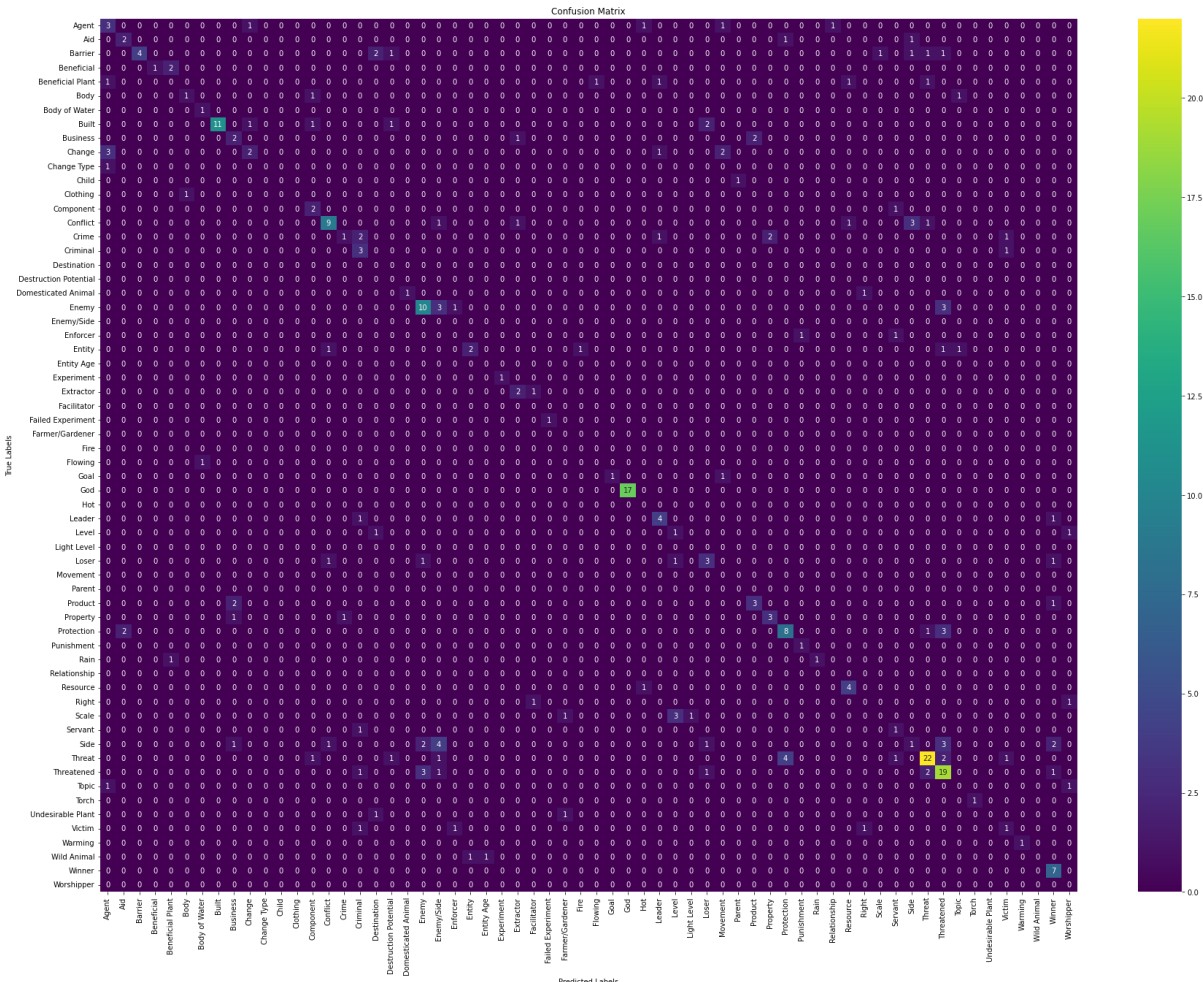

Figure 3: Confusion matrix of test outcomes of the single-task baseline to predict highlighted aspects. No literal metaphor provided in the input.

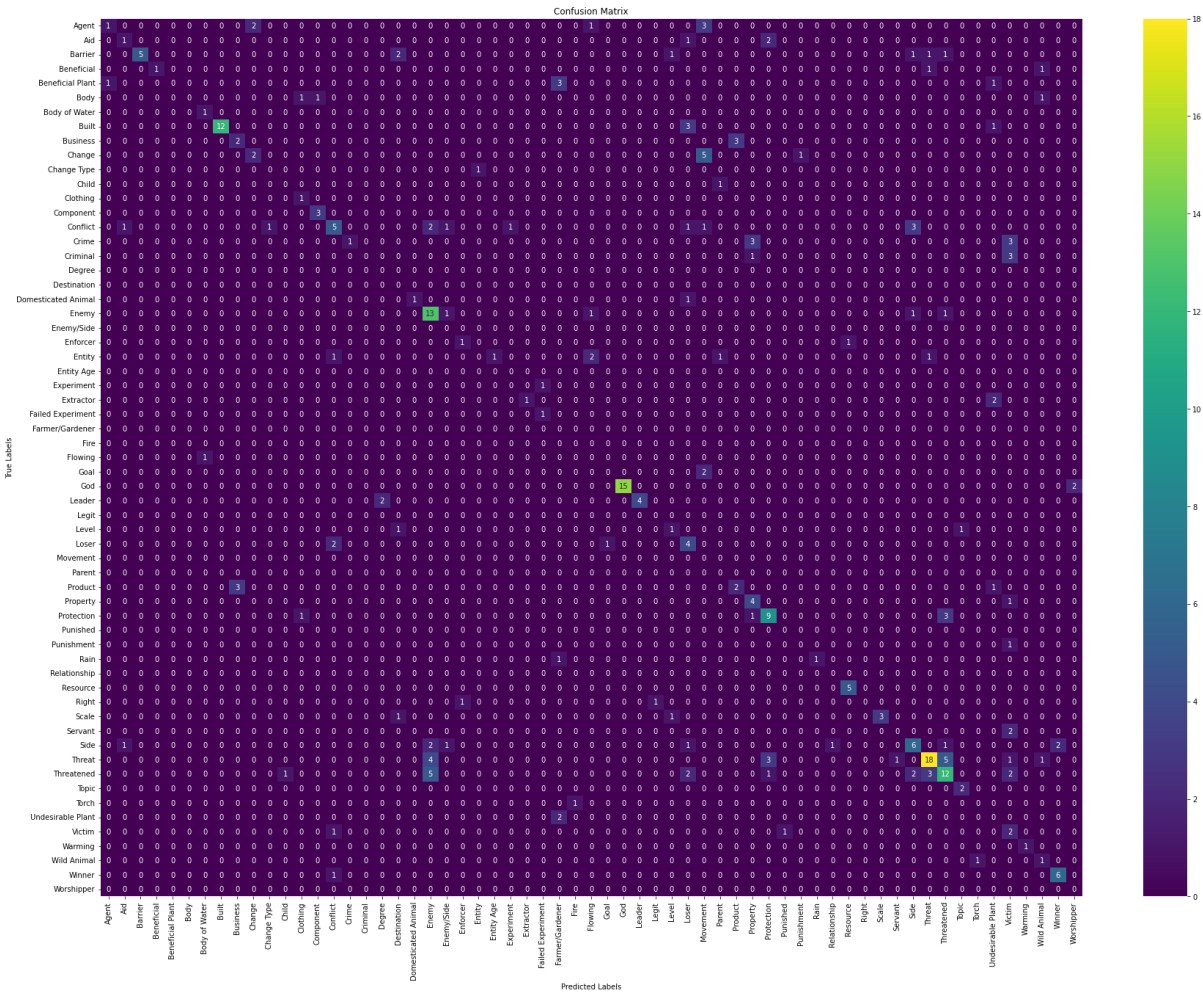

Figure 4: Confusion matrix of test outcomes of the joint learning approach to predict highlighted aspects. No literal metaphor provided in the input.

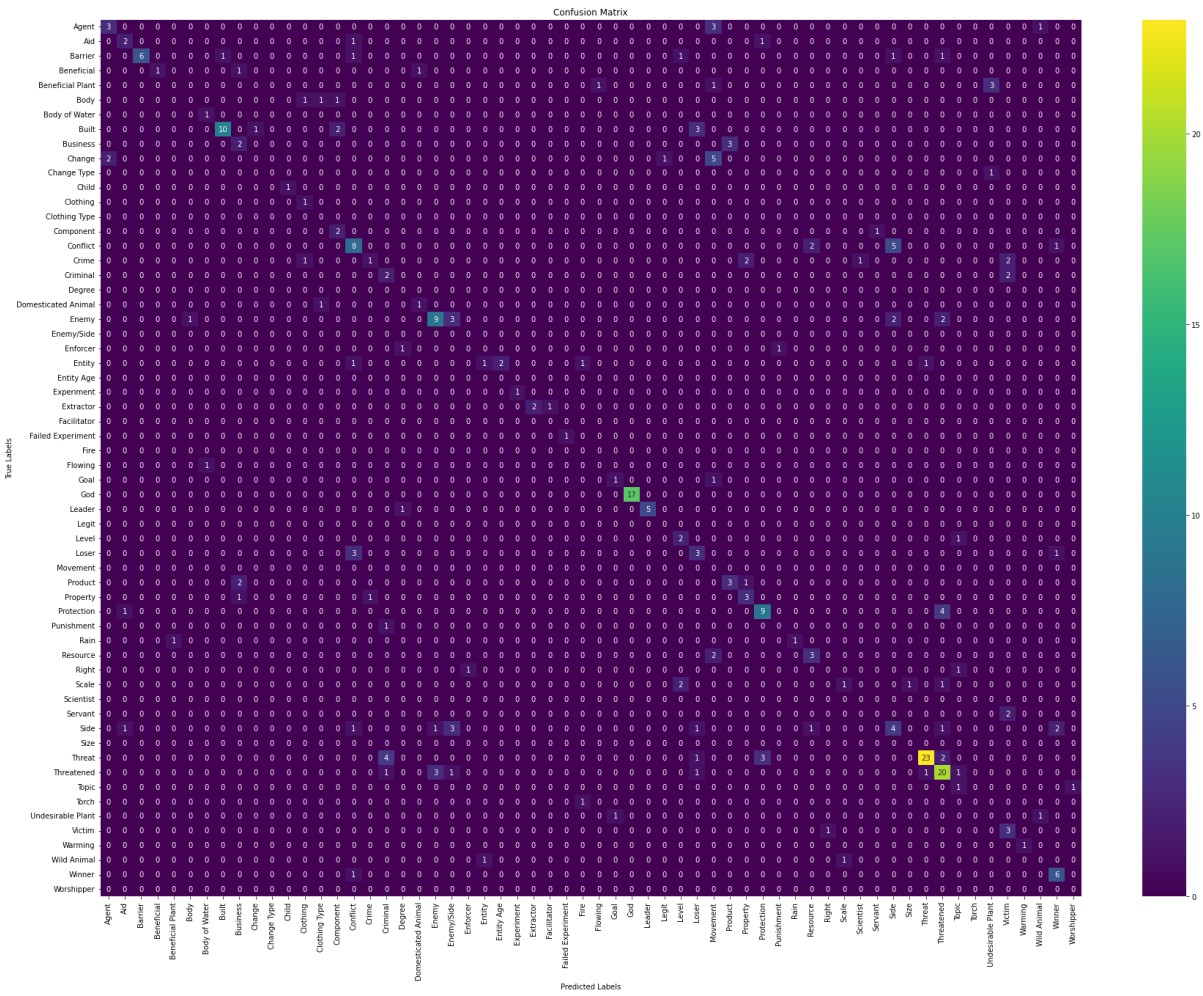

Figure 5: Confusion matrix of test outcomes of the continual learning approach to predict highlighted aspects. No literal metaphor provided in the input.

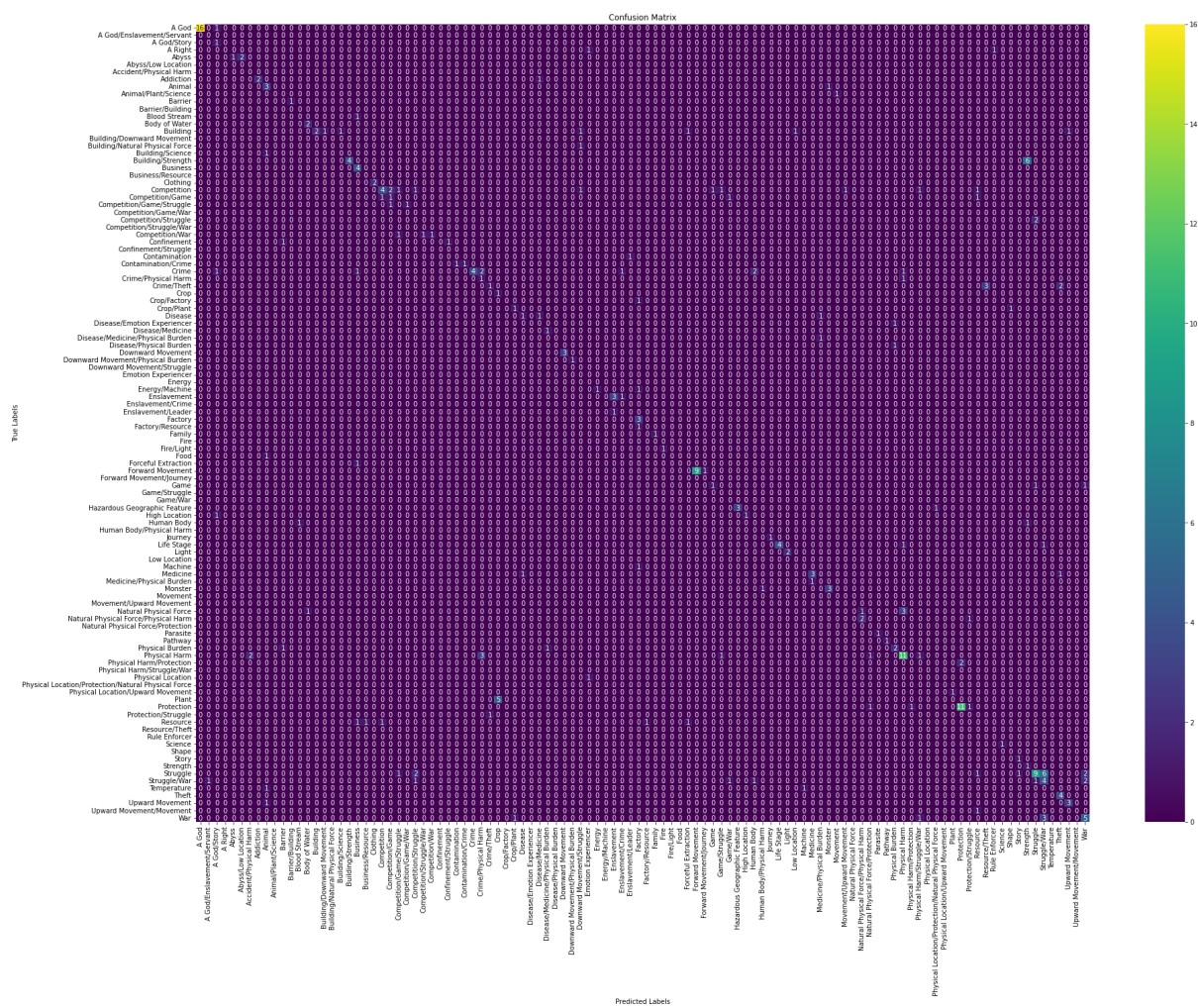

Figure 6: Confusion matrix of test outcomes of the single-task baseline to predict source domains. No literal metaphor provided in the input.

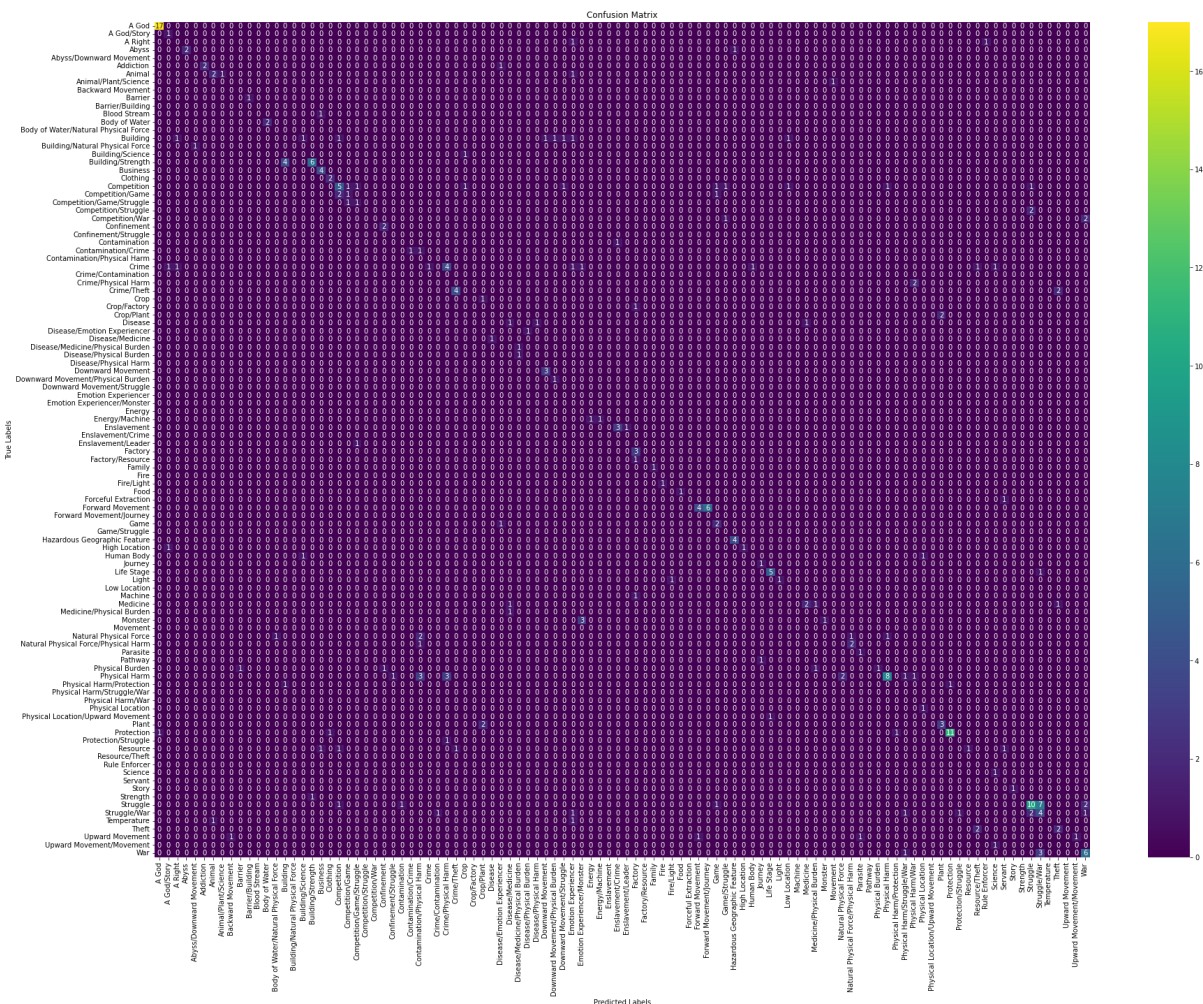

Figure 7: Confusion matrix of test outcomes of the joint learning approach to predict source domains. No literal metaphor provided in the input.

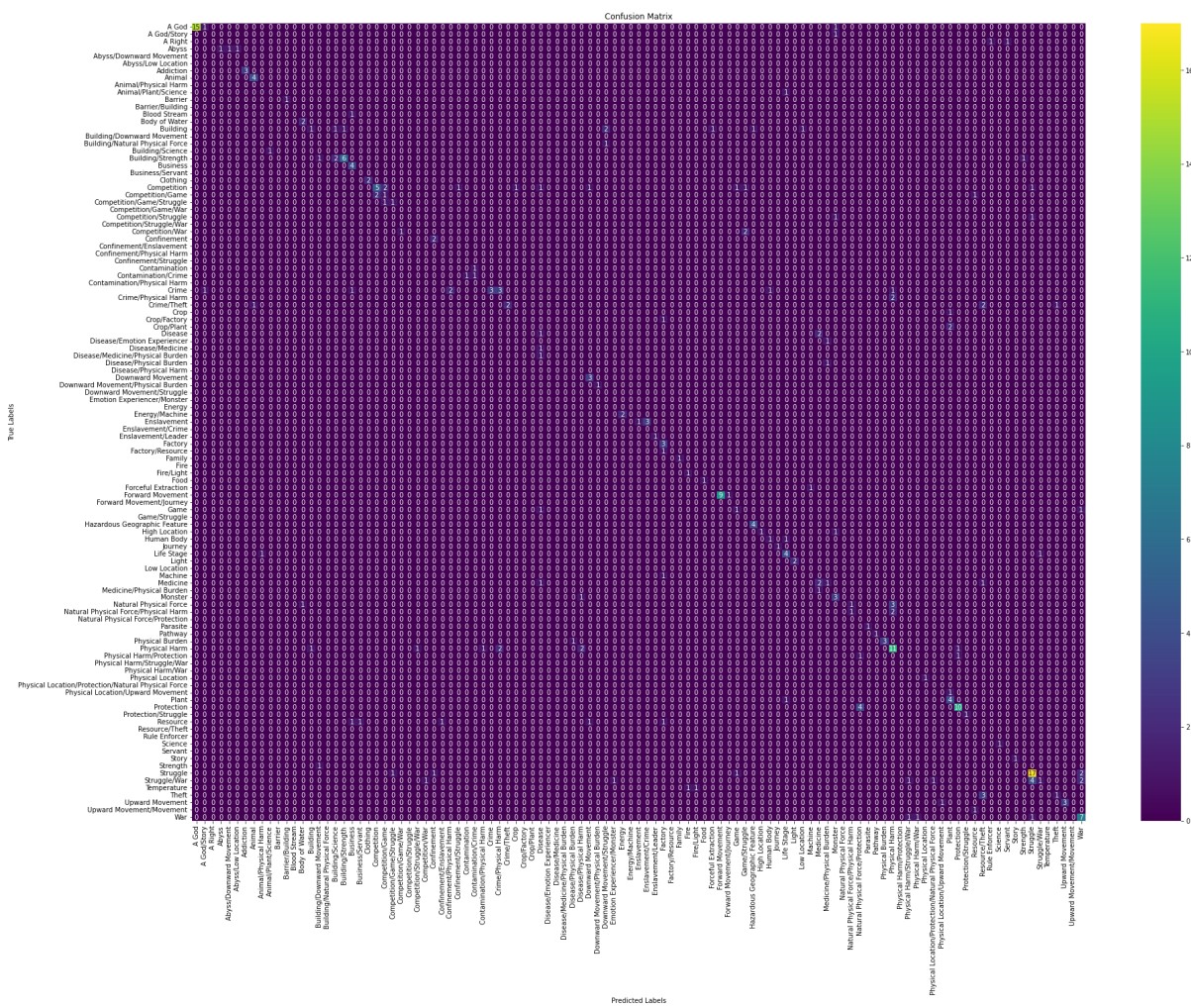

Figure 8: Confusion matrix of test outcomes of the continual learning approach to predict source domains. No literal metaphor provided in the input.

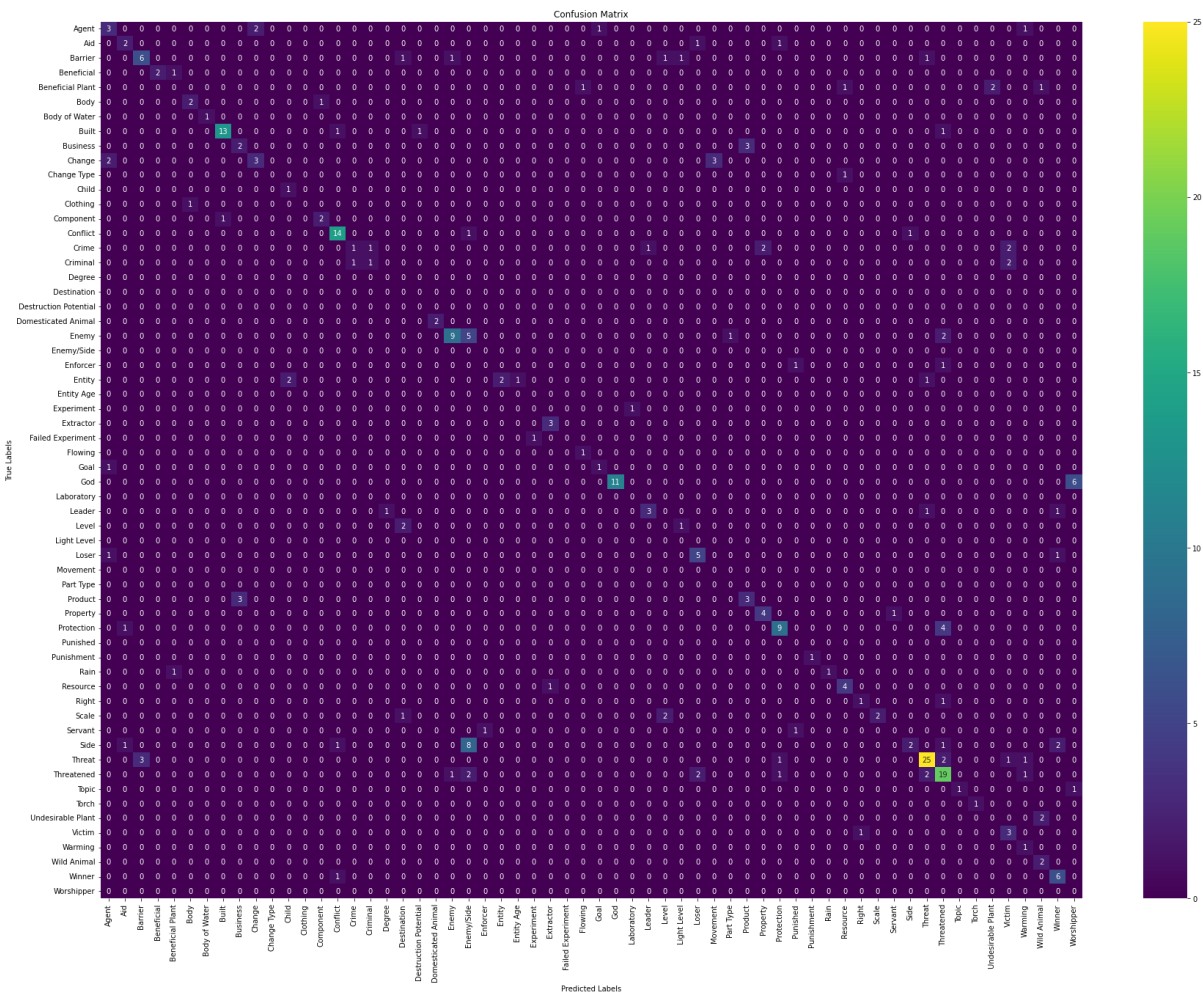

Figure 9: Confusion matrix of test outcomes of the single-task baseline to predict highlighted aspects. Literal metaphor provided in the input.

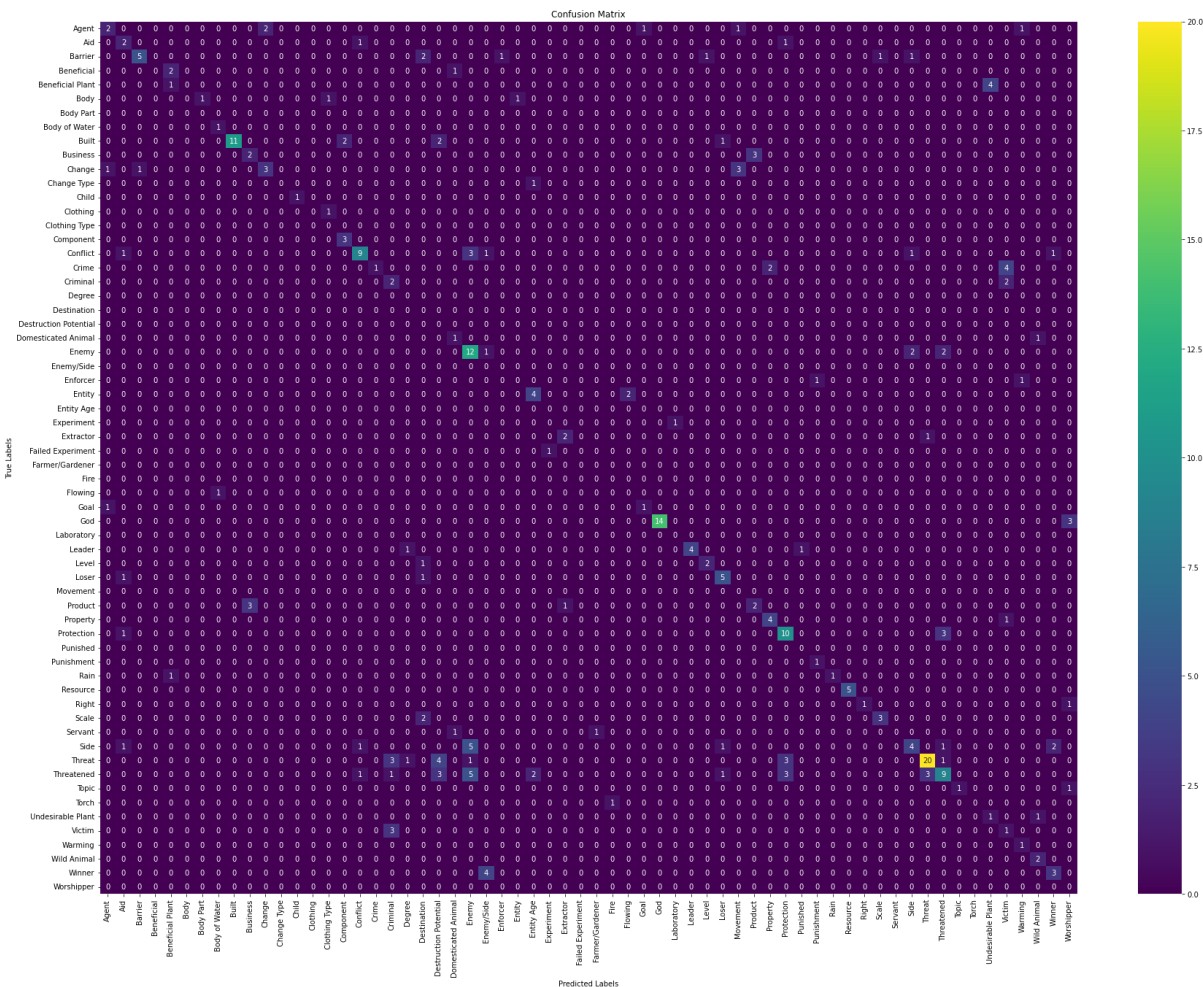

Figure 10: Confusion matrix of test outcomes of the joint learning approach to predict highlighted aspects. Literal metaphor provided in the input.

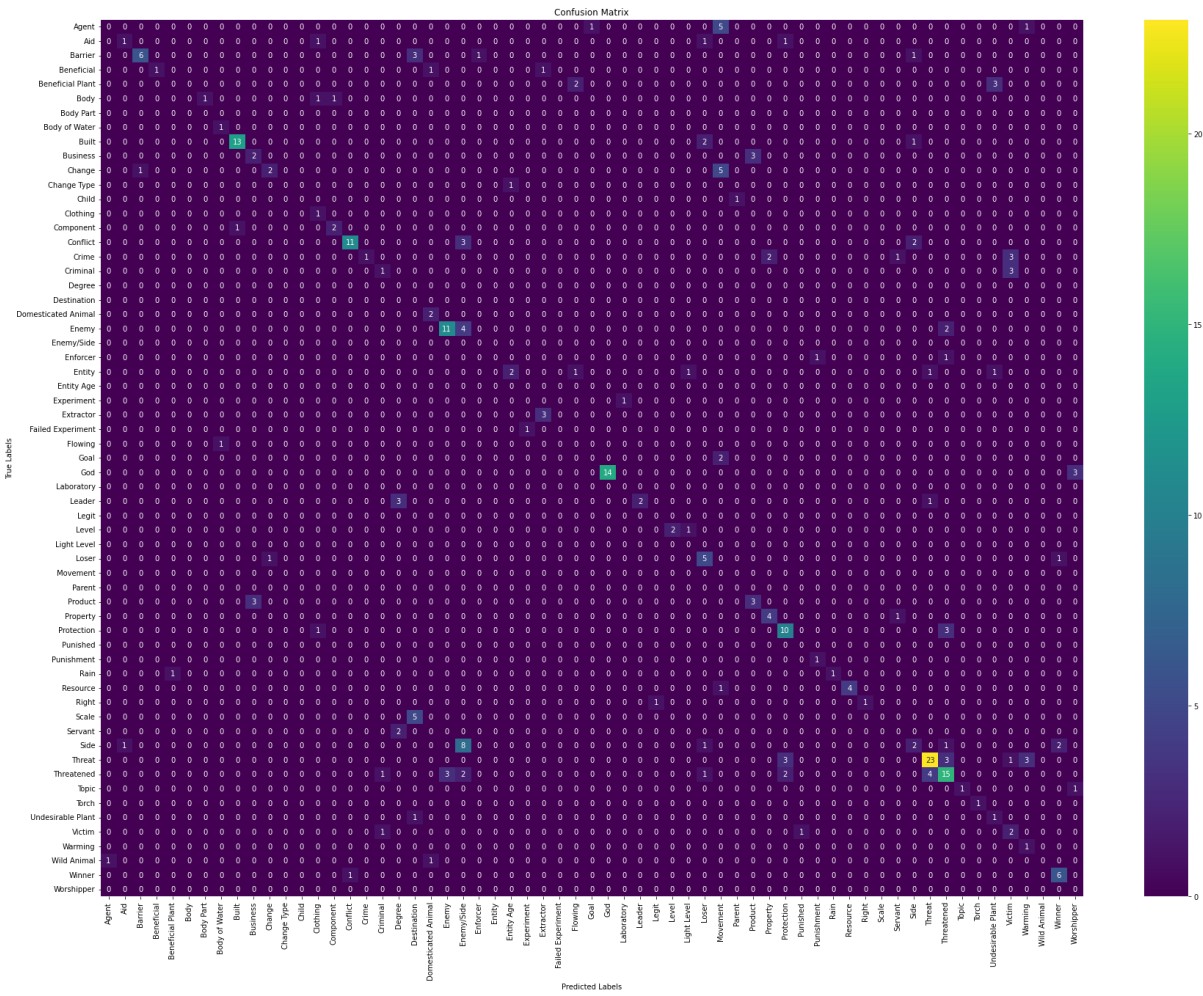

Figure 11: Confusion matrix of test outcomes of the continual learning approach to predict highlighted aspects. Literal metaphor provided in the input.

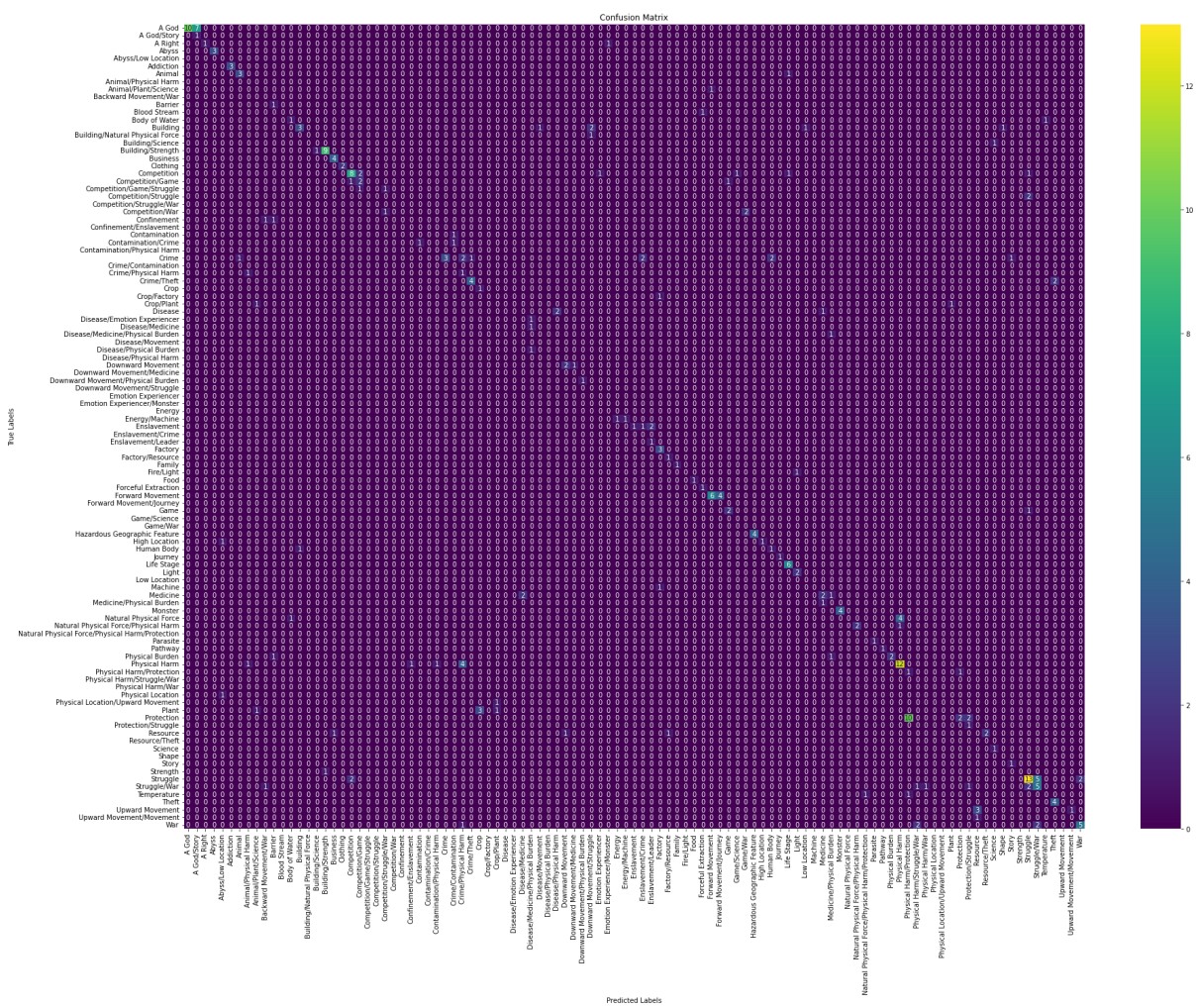

Figure 12: Confusion matrix of test outcomes of the single-task baseline to predict source domains. Literal metaphor provided in the input.

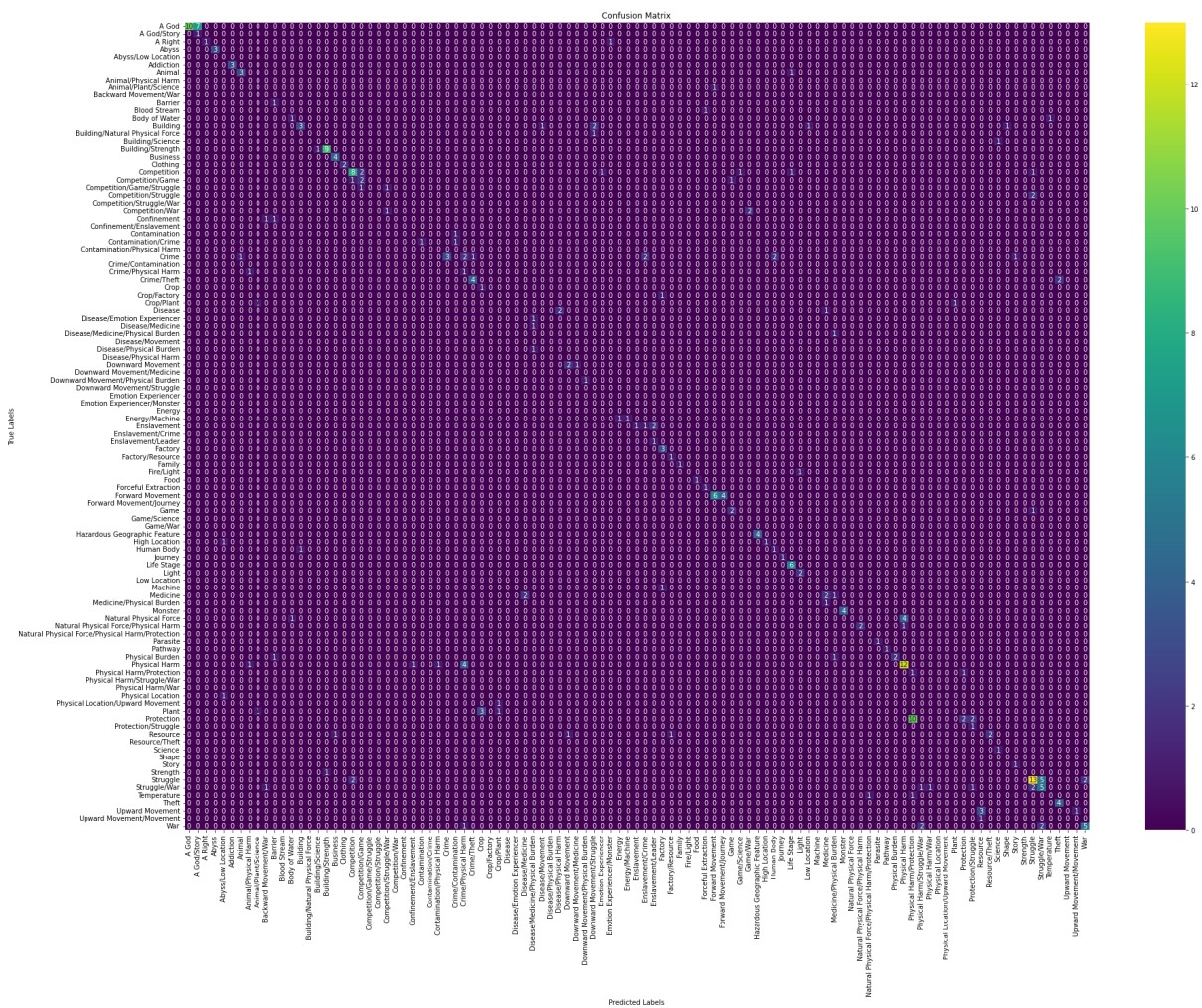

Figure 13: Confusion matrix of test outcomes of the joint learning approach to predict source domains. Literal metaphor provided in the input.

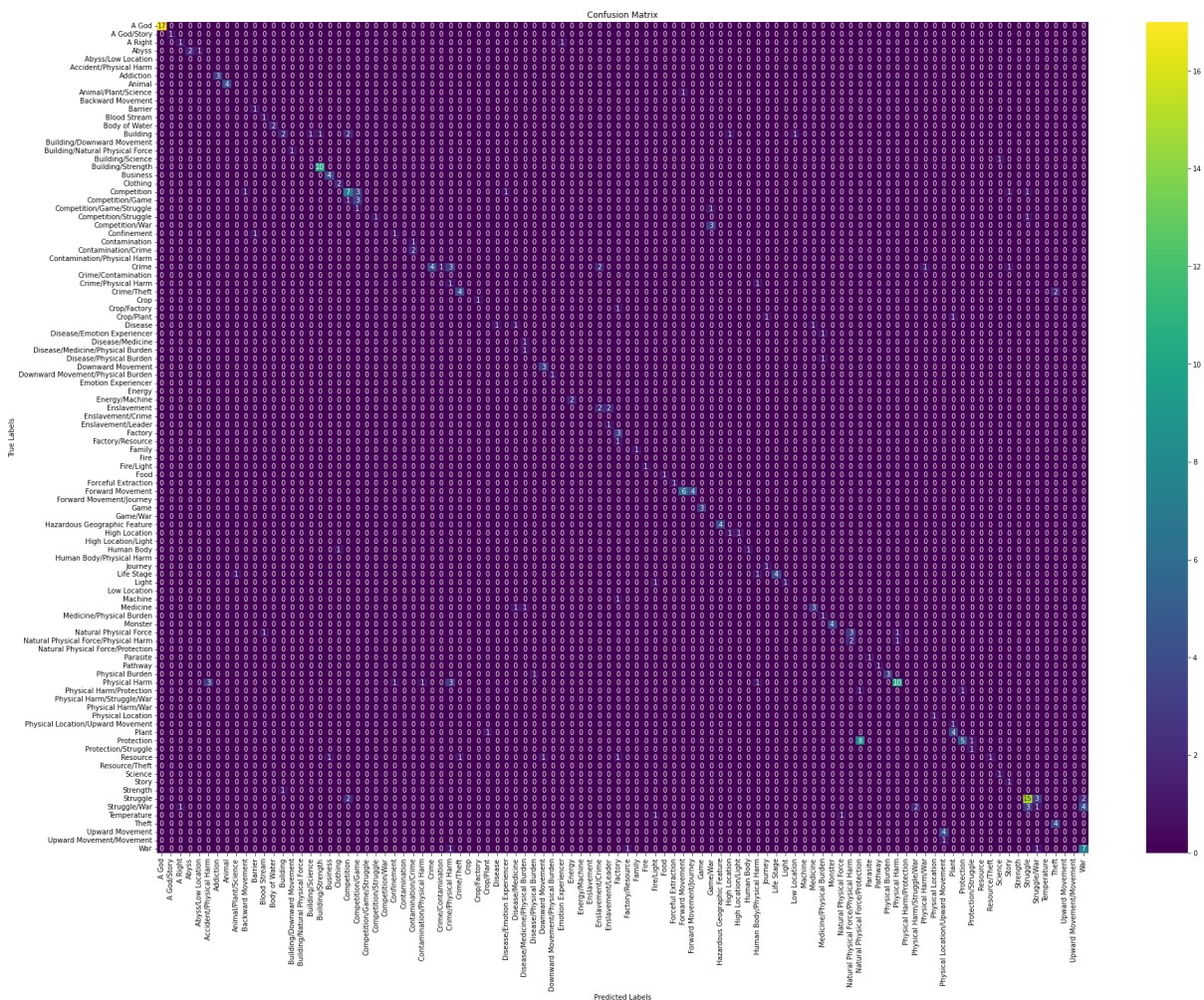

Figure 14: Confusion matrix of test outcomes of the continual learning approach to predict source domains. Literal metaphor provided in the input.