# OpenReview forum: "Modeling Highlighting of Metaphors in Multitask Contrastive Learning Paradigms"
_EMNLP/2023/Conference — EMNLP 2023 Findings_

### Official Review · Reviewer_u9Nz · 2023-08-04

**Soundness:** 4

**Excitement:**

4: Strong: This paper deepens the understanding of some phenomenon or lowers the barriers to an existing research direction.

**Paper Topic And Main Contributions:**

This paper proposes a novel contrastive learning method for metaphor interpretation. In particular, this work introduces a new task for predicting the highlighted aspects in metaphors. This proposed task takes advantage of various linguistic research on metaphor, such that metaphor is to highlight certain aspects of its target domain. Such a task has potential in both NLP and linguistic research. The authors propose to use the multitask learning scheme to jointly predict the source domains and the highlighted aspects.

**Reasons To Accept:**

1) The newly proposed task is both novel and useful.

2) Good writing.

3) Detailed analysis of experimental results.

4) Good conclusions, especially the "limitations" section.

**Reasons To Reject:**

1) The paper has two main contributions. The first contribution is kicking off a new task, and the second contribution is proposing a multi-task learning approach to accomplish this task. The second of these contributions mainly follows an ideological approach that has been used for a long time in the NLP community, and the main innovativeness is focused on the first contribution. However, the analysis part of the experiment is discussed mainly for the second contribution. From my perspective, it's better to focus more on the first contribution instead of the second one.

2) There is only 1 baseline model in the experimental section.

**Reproducibility:**

5: Could easily reproduce the results.

**Reviewer Confidence:**

4: Quite sure. I tried to check the important points carefully. It's unlikely, though conceivable, that I missed something that should affect my ratings.

---

> ### Author Rebuttal · Authors · 2023-08-28
>
> Thank you for your encouraging assessment of the paper in terms of the presented task, analyses, and discussion. Regarding the concerns you raised:
>
> **The first contribution is kicking off a new task, and the second contribution is proposing a multi-task learning approach to accomplish this task. The second of these contributions mainly follows an ideological approach that has been used for a long time in the NLP community, and the main innovativeness is focused on the first contribution. However, the analysis part of the experiment is discussed mainly for the second contribution. From my perspective, it's better to focus more on the first contribution instead of the second one.**
>
> We understand your view and shall elaborate further on the first contribution in the revised version of the paper.
>
> **There is only 1 baseline model in the experimental section.**
>
> As indicated in the response to Reviewer hYfy already, the reason for having a single-task baseline only is essentially because the task is new and we consider the only other relevant work as stated in the paper. By resorting to MTL, we investigated how the knowledge of one (e.g., source domains) influences the prediction of the other (e.g., highlighted aspects).

---

### Official Review · Reviewer_hYfy · 2023-08-04

**Typos Grammar Style And Presentation Improvements:** "a principally model"
**Soundness:** 4

**Excitement:**

2: Mediocre: This paper makes marginal contributions (vs non-contemporaneous work), so I would rather not see it in the conference.

**Paper Topic And Main Contributions:**

This paper presents an analysis of metaphor detection with the use of source domain and the aspect that is highlighted in the text. The authors experiment with two multi-task learning training scenarios and show that this joint baseline outperforms the a single-task baseline.

**Reasons To Accept:**

This is an interesting task and deepens our understanding of automatic approaches to metaphor.

There is a deep and interesting qualitative analysis of the results.

The paper is well-written and clear.

**Reasons To Reject:**

The focus of this paper is somewhat narrow in that it depends on a corpus with very specific annotations and there is no wider comparison. In particular, it would be interesting to see if this work could be used to improve a the more general case of annotation.

The only baseline is a single-task baseline, and as the MTL set-up is not really novel, the results are not surprising (as it has been well-established that MTL improves peformance).

This is a nice paper, but I feel it would be better suited to a focused workshop rather than the EMNLP main conference

**Reproducibility:**

4: Could mostly reproduce the results, but there may be some variation because of sample variance or minor variations in their interpretation of the protocol or method.

**Reviewer Confidence:**

3: Pretty sure, but there's a chance I missed something. Although I have a good feel for this area in general, I did not carefully check the paper's details, e.g., the math, experimental design, or novelty.

---

> ### Author Rebuttal · Authors · 2023-08-28
>
> Thank you for the positive evaluation of the task and insights provided, but also for sharing your concerns.
>
> **The focus of this paper is somewhat narrow in that it depends on a corpus with very specific annotations**
>
> We point out that the annotations are needed for training and evaluation only, while our approach can be applied to any metaphorical sentence.
>
> **there is no wider comparison. In particular, it would be interesting to see if this work could be used to improve a the more general case of annotation.**
>
> Our broader focus is the interpretation of metaphor as a figurative language as stated in lines 039-051 and 072-080. In order to do that, we have proposed the novel task to model highlighted aspects - a phenomenon of metaphorical meaning construction that was never addressed before although being popular in theoretical linguistics (Lakoff and Johnson, 2003). We consider this to be a starting point, based on which further research can be done in order to expand this to figurative languages even beyond metaphors like sarcasm, or idioms.
>
> **The only baseline is a single-task baseline**
>
> The reason for having a single-task baseline only is essentially because the task is new and we consider the only other relevant work as stated in the paper. By resorting to MTL, we investigated how the knowledge of one (e.g., source domains) influences the prediction of the other (e.g., highlighted aspects).
>
> **as the MTL set-up is not really novel, the results are not surprising (as it has been well-established that MTL improves peformance).**
>
> Conceptually, we agree: Designing the task a multi-task paradigm intuitively makes sense because of the aforementioned reason. Practically, multi-task learning does perform well in certain cases, but it also does not do so throughout (e.g., the joint learning approach does not perform as well as continual learning). In our opinion, the novelty lies in the combination of the new task (as acknowledged by all reviewers) and successively devising an approach that technically utilizes the information in this setting of metaphorical interpretation.

---

### Official Review · Reviewer_onsm · 2023-08-04

**Soundness:** 3

**Excitement:**

2: Mediocre: This paper makes marginal contributions (vs non-contemporaneous work), so I would rather not see it in the conference.

**Paper Topic And Main Contributions:**

Metaphorical words transfer from a source to a target domain and Sengupta et al. (2022) studied contrastive learning for predicting the source domain from a metaphorical sentence. This study extended the work of Sengupta et al. by focusing also on the metaphorical word or text span (aspect). The authors used a baseline a single-task model, predicting either the aspect or the source domain. They developed two multi-task approaches, one based on continual and one based on joint learning. Small differences in accuracy were found, compared to the baseline, which are not enough to draw solid conclusions, at least not with out any statistical significance measurement.


**Questions For The Authors:**

A. What are the predictions of the texts presented in L560 without the addition of the literal metaphor to the input?

**Reasons To Accept:**

* A new task is proposed, that of better understanding the metaphorical text span in a metaphorical sentence.
* The comparison between single and multi task learning is interesting, and potentially reaching a wider audience.


**Reasons To Reject:**

* The results are missing statistical significance while the differences in scores are very small.
* A single dataset was used for experiments, and, judging by the annotations presented in this paper, the inter-annotator agreement should also be discussed.


**Reproducibility:**

3: Could reproduce the results with some difficulty. The settings of parameters are underspecified or subjectively determined; the training/evaluation data are not widely available.

**Reviewer Confidence:**

3: Pretty sure, but there's a chance I missed something. Although I have a good feel for this area in general, I did not carefully check the paper's details, e.g., the math, experimental design, or novelty.

**Typos Grammar Style And Presentation Improvements:**

- L88: The hypothesis the authors refer to is not clear to me.
- Figure 1 (caption): L96-97 writes "or" which is sort of contradicting.
- L89: The contribution in proposing these two multitask setups is not very clear.
- L157: Please consider adding a definition of a literal metaphor.
- L173: Based on the contributions, the goal is to predict both the aspect and the source domain, but now it's stated differently. This makes the presentation confusing. Also, my understanding is that you do not predict the aspect, but its domain or topic (predicting the aspect would be to predict the word/term).
- L242: Not clear what is this embedding score, so I am guessing some sort of distance or similarity.
- L255: Not clear what are these concepts and how are they represented exactly.
- L268: Not clear how does backpropagation relates to this specific loss computation. Some re-wording would make this clear.
- L276: What previous approach, the previous work (L181) or continual learning?
- L300: The exact type of combination is not shared.
- L332: Where is the difficulty in the real world? (Also, which information?)
- L351-358: Do you mean that the label is predicted as a next token, with its probability compared to those of alternative labels?
- L400-403: Not the right conclusion, in my opinion. Without the literal metaphor, in Acc-3, it performs similarly to the baseline for highlights and source; 0.01 up in Acc-1 for highlights; better in Acc-5 for source. With the literal metaphor, the baseline is better in 3 out 6 columns.
- Table 2: In acc@3 of aspects without the metaphor, the baseline is equally well but not in bold.
- L474: In my eyes, both are correct while "destruction potential" sounds better. Please consider elaborating more regarding your perspective on what's right and wrong.
- L529: The differences reported do not look considerable to me.
- L544: Perhaps you can elaborate more on why was this unexpected.
- L610-612: I see a different pattern. Relaxation reduces the difference in acc@1 between continual learning and the single-task baseline, from 0.011 to 0.007, for highlights, and from 0.034 to 0.014 for source.
- Table 4: What is presented in bold? (Two per row and two per column...)

---

> ### Author Rebuttal · Authors · 2023-08-28
>
> Thank you for your positive judgment about the proposed task and the insights from the multitask learning experiments as well as for your detailed feedback.
>
> **What are the predictions of the texts presented in L560 without the addition of the literal metaphor to the input?**
>
> Files containing all predictions of all experiments have been submitted with the code, found in the folder /project_metaphor/analysis/ with the files starting with 'confmtrx-' to be the analysis files.
>
> The predictions of the texts without the literal metaphor added to the input in continual learning comes as per our expectation - where for all three examples from the table, our approach predicted the highlighted aspect correctly - supporting our intuition in the paper (concerning line 560). By literal metaphor, we mean the metaphorical expression, as found in the text; we will clarify this.
>
> **The results are missing statistical significance while the differences in scores are very small.**
>
> We will add significance results when we revise the paper. Indeed, the smaller differences may not be significant.
>
> **A single dataset was used for experiments**
>
> Our task design investigates how source domains can influence prediction of highlighted aspects and vice versa. As discussed in lines 280-282, to the best of our knowledge, the dataset we used is the only one with annotations of source domains and highlighted aspects.
>
> **judging by the annotations presented in this paper, the inter-annotator agreement should also be discussed.**
>
> Since the corpus comes from peer-reviewed work (Gordon et al., 2015), we did not explicitly question the reliability of the annotations. Gordon et al. reported Cohen's kappa agreement values between 0.42 and 0.65. While they regard these kappa values as good given the complexity of the task, we agree that they might pose a problem. However, this is the only dataset that we can use. Furthermore, it is unlikely that the lack of agreement leads to systematically biased conclusions on our side, but rather that it adds noise.

---

### Meta-Review · Area_Chair_4JEC · 2023-09-12

**Recommendation:** 3

**Metareview:**

The paper focuses on examining highlighting in metaphors, where certain aspects of a target domain are emphasized by the use of the metaphor.  The authors introduce a new NLP task of identifying the highlighted aspect in a metaphorical sentence.  Leveraging a source-target domain framework, the paper explores whether jointly predicting the highlighted aspect and the source domain in a metaphorical sentence improves predictive performance on both tasks.  They find that by performing continual learning, they are able to obtain performance improvements on both source domain and highlight prediction by allowing information to be shared.

On the whole, the reviewers thought the paper was sound and well-written.  One of the main issues in terms of soundness was brought up by reviewer onsm, who noted that the results lacked statistical significance tests despite the different models having similar accuracies across tasks.  The authors discussed including significance tests if they revise the paper but mentioned that their results may no longer be significant after testing.  This, along with the fact that the authors only compare their approach with a simple single-task baseline (noted by reviewers hYfy and u9Nz), does weaken the conclusions the paper draws on jointly predicting source domain and highlighting.  Another soundness critique by reviewer onsm is that the inter-annotator agreement for the dataset used was not acknowledged.  As stated in the rebuttal, agreement on the dataset is quite low, ranging from 0.42 to 0.65, suggesting there may be issues with the dataset used.  However, given that the dataset used is from Gordon et al. (2015), the interannotator agreement does not appear to be an explicit soundness issue with this paper directly.  The paper, however, could be strengthened by acknowledging issues with the single dataset used in the limitations section.

On the other hand, while the paper was well-received in terms of soundness, there is some disagreement over how substantial a contribution it made with its multitask experiments.  While reviewer onsm appreciates that the paper runs experiments examining the difference between single and multitask learning for a metaphor interpretation task, reviewers hYfy and u9Nz regard that comparison as a weakness of the paper.  As reviewer u9Nz notes, the multitask learning experiment “mainly follows an ideological approach that has been used for a long time in the NLP community”.  As a result, the methodological approaches and results in the paper are not surprising, despite operating in a new task setup.

Overall, while the paper is well-written and most of the reported technical details in the experiments are sound, there remain questions about how exciting of a contribution the paper makes.  This is especially the case, given that many of the issues in the paper are due to the limited availability of existing high-quality datasets for metaphor highlighting.  Thus, work that can be done to follow up on the observations and analyses in this paper may be limited.

---

### Decision · Program_Chairs · 2023-10-07

**Decision:**

Accept-Findings

**Comment:**

The paper focuses on examining highlighting in metaphors, where certain aspects of a target domain are emphasized by the use of the metaphor.  The authors introduce a new NLP task of identifying the highlighted aspect in a metaphorical sentence.  Leveraging a source-target domain framework, the paper explores whether jointly predicting the highlighted aspect and the source domain in a metaphorical sentence improves predictive performance on both tasks.  They find that by performing continual learning, they are able to obtain performance improvements on both source domain and highlight prediction by allowing information to be shared.

On the whole, the reviewers thought the paper was sound and well-written.  One of the main issues in terms of soundness was brought up by reviewer onsm, who noted that the results lacked statistical significance tests despite the different models having similar accuracies across tasks.  The authors discussed including significance tests if they revise the paper but mentioned that their results may no longer be significant after testing.  This, along with the fact that the authors only compare their approach with a simple single-task baseline (noted by reviewers hYfy and u9Nz), does weaken the conclusions the paper draws on jointly predicting source domain and highlighting.  Another soundness critique by reviewer onsm is that the inter-annotator agreement for the dataset used was not acknowledged.  As stated in the rebuttal, agreement on the dataset is quite low, ranging from 0.42 to 0.65, suggesting there may be issues with the dataset used.  However, given that the dataset used is from Gordon et al. (2015), the interannotator agreement does not appear to be an explicit soundness issue with this paper directly.  The paper, however, could be strengthened by acknowledging issues with the single dataset used in the limitations section.

On the other hand, while the paper was well-received in terms of soundness, there is some disagreement over how substantial a contribution it made with its multitask experiments.  While reviewer onsm appreciates that the paper runs experiments examining the difference between single and multitask learning for a metaphor interpretation task, reviewers hYfy and u9Nz regard that comparison as a weakness of the paper.  As reviewer u9Nz notes, the multitask learning experiment “mainly follows an ideological approach that has been used for a long time in the NLP community”.  As a result, the methodological approaches and results in the paper are not surprising, despite operating in a new task setup.

Overall, while the paper is well-written and most of the reported technical details in the experiments are sound, there remain questions about how exciting of a contribution the paper makes.  This is especially the case, given that many of the issues in the paper are due to the limited availability of existing high-quality datasets for metaphor highlighting.  Thus, work that can be done to follow up on the observations and analyses in this paper may be limited.